# Evaluation of the Impact of VR Rural Streetscape Enhancement on Relaxation–Arousal Responses Based on EEG

**Hongguo Ren** [1], **Yujun Wang** [1], **Jing Zhang** [1,*], **Ziming Zheng** [1] **and Qingqin Wang** [2]

1   International Research Center of Architecture and Emotion, Hebei University of Engineering, Handan 056038, China; renhongguo@hebeu.edu.cn (H.R.); 15312889667@163.com (Y.W.); bonjourlapluie@outlook.com (Z.Z.)
2   China Academy of Building Research, Beijing 100013, China
*   Correspondence: zhangjing01@hebeu.edu.cn; Tel.: +86-186-5920-1726

**Abstract:** As the quality of life and the spiritual and cultural well-being of the inhabitants progress, the current rural infrastructure has challenges in adequately addressing the physical and psychological requirements of individuals. This work presents a method for evaluating rural habitats by utilizing electroencephalography (EEG) signals and virtual reality (VR) technology to address the existing gap in physiological data evaluation in rural areas. We choose as experimental images the current situation (C1–C5) scenes of five rural main street nodes as well as the comparative transformation scenes (T1–T5). It then assesses the subjects' subjective cognition and level of relaxation–arousal responses using the $\alpha/\beta$ value in the EEG data and the three subjective scale indexes of "Interest", "Comfort", and "Vitality". The study's findings demonstrated the following: 1. All three scores increased in the transformation scenarios, and subjects' subjective assessments varied significantly across all five sets of scenes. 2. In all $\alpha/\beta$ values where significant differences existed, every electrode demonstrated a relaxation response in the transformation scenes (T1–T5) compared to the current scenes (C1–C5), whereas the T8 electrode demonstrated the only arousal response. 3. The Pz electrode in the parietal lobe area was found to be the most sensitive to the visual response to the green landscape and the form of the building façade along the street, and the T8 electrode in the right temporal lobe area was the most sensitive to the response to the overall perception of the surrounding environment, according to a comparison of the longitudinal $\alpha/\beta$ value. More options for rural streetscape design as well as fresh insights and methodologies for assessing the rural human environment in the future are anticipated from this study.

**Keywords:** EEG; VR; rural habitat; rural streetscapes; relaxation–arousal response

## 1. Introduction

### 1.1. Research Background

Improving rural habitat helps foster beneficial interactions between the living environment and the rural social, economic, and resource environments. It is a significant index of the resident's quality of life. Improving rural habitat has become a key initiative to realizing the rural revitalization strategy [1]. To analyze the current environmental status and lead the improvement measures, a scientific and reliable evaluation technique must be established. Most of the research that has already been conducted is based on the three dimensions of environment, society, and economy. These studies selected the appropriate index factors, built an indicator evaluation system [2] for the rural habitat environment, established a multi-criterion decision-making model [3,4], and objectively assessed the quality of the sustainable development level and the rural habitat environment using the entropy weight method [3], hierarchical analysis [5], principal component analysis [6], and GIS technology platform [7]. Furthermore, studies on the perceptions of rural residents are steadily growing to investigate the satisfaction of the rural habitat environment, develop

a subjective evaluation scale and evaluation system, and learn more about the villagers' sense of participation and participation behavior [8,9].

Index assessment methods, which are widely used as the predominant evaluation approach in contemporary research on rural human settlements, are not without their limitations. The comprehensive analysis of complex environmental phenomena may be limited in its ability to encompass all aspects, potentially overlooking certain factors that are challenging to quantify, such as cultural values, emotions, and subjective feelings. Additionally, the subjective and controversial nature of assigning importance and weight to different indexes may result in evaluation outcomes that lack sufficient accuracy. Furthermore, the subjective rating scale exhibits a deficiency in accuracy attributable to the overall limited cognitive abilities of the inhabitants and the inherent bias introduced by the structure of the questionnaire, hence leading to a dearth of precision in the obtained questionnaire outcomes.

To this day, there has been a limited number of research that has explicitly assessed the rural habitat landscape in terms of its impact on human physiology. Electroencephalography (EEG) can serve as an objective measure of physiological responses to assess individuals' holistic experience of their immediate surroundings. Hence, this study focuses on the rural streetscape, a significant component of the rural habitat environment. The objective was to investigate the impact of rural streetscape on individual environmental perceptions and emotional responses using electroencephalogram (EEG) measurements within an immersive virtual reality (VR) setting. Additionally, the study aimed to evaluate people's satisfaction with the design of enhanced rural streetscapes, thereby offering novel insights and approaches for assessing the landscape of the rural habitat environment.

*1.2. Synthesis of Relevant Studies*

Our perception of emotional, physiological, and cognitive function was influenced by the built environment [10,11], and design elements including shape, layout, materials, scale, and color have been found to correlate with emotional reactions in neuro-scientific experiments [12]. Curvilinear formal spaces resulted in more joyful experiences [13]; natural materials lower stress and anxiety and increase physiological and cognitive function [14]. Grey and vibrantly colored interior spaces were conducive to work and pleasure [15].

Streetscapes, one of the most prevalent forms of built environments, were places where many people congregate and where their functional diversity, façade design, and scale characteristics impacted how people perceive and use the space [16]. Research has indicated that elements of the streetscape enclosure, such as the number of buildings on the block, cross-sectional dimensions, and street tree canopies, positively influenced people's sense of safety [17]. The visual and physiological comfort of users was greatly influenced by high levels of greenery and water, low building façade densities, and moderately complex street configurations [18]. People also favored older buildings whose external design was based on more traditional compositional patterns, and the activities of buildings also significantly affected perception [19].

Related fields of study included the rural environment. People's emotions and actions were particularly influenced by their perceptions of rural settings. From the viewpoint of the locals, M. Campos [20] investigated awareness and impressions of rural landscapes. According to Jamal-e-Din Mahdi Nejad [21], rural landscapes' aesthetics, readability, clarity, and reading environment were extremely important, and their perception was also much influenced by symbols, landscapes, and shapes in the subjective index. Yanlong Guo [22] assessed the four aspects of rural landscape design—rural landscape ecology, water environment, climate, and sound. He concluded that the most important factors influencing adolescents' perceptions and experiences of the countryside were sound comfort, air cleanliness, and landscape adaptability. Simultaneously, the enhancement of flora also played a role in improving the visual perceptual experience of the rural environment among young individuals.

Neuroscience is gradually becoming a powerful tool for exploring the field of built environments, as evidence suggests that human perception of streetscapes may be closely related to patterns of brain activity. This is one way that the built environment is influencing human perception. Within the realm of research methodologies, EEG has emerged as a prominent technical instrument for investigating individuals' perceptions of design elements and their subjective preferences for the constructed environment. Additionally, VR has recently been employed in environmental perception tests, further expanding the scope of study in this domain [23,24]. Sara Tilley [25] employed a combination of EEG and interviews as a research methodology to gain insights into the subjective experiences of older adults in various urban settings. The study revealed that older participants exhibited varying levels of "arousal", "engagement", and "frustration" during their transitions between bustling city streets and urban green spaces. Tian Gao [26] employed VR technology in a study aimed at examining the physiological (EEG) and psychological (attention, positive emotions, negative emotions) reactions and individual preferences in various urban settings. The findings revealed that partially open green spaces elicited the most significant positive impact on negative emotions, whereas closed green spaces were associated with the least recreational preference. Sanghee Kim [27] employed the integration of EEG and VR to quantitatively assess the influence of modifications in architectural features on individuals' emotional states. Kim proposed specific combinations of ceiling heights and window proportions that were shown to elicit the least amount of arousal in users. It followed that the integration of EEG and architecture had the potential to enhance human and societal well-being by evaluating the effectiveness of the current built environment and making more focused design choices [28].

EEG waves are commonly classified into distinct frequency ranges, namely alpha, beta, gamma, theta, and delta waves. Alpha ($\alpha$) waves have a frequency range of 8–13 Hz, which is indicative of states of relaxation, comfort, and enjoyment. On the other hand, Beta ($\beta$) waves are characterized by a frequency range of 14–30 Hz, suggesting heightened cognitive engagement, positive arousal, and focused attention. The relaxation arousal response is predominantly linked to alpha and beta brain waves.

Tee et al. [29] demonstrated a negative correlation between the $\alpha$ to $\beta$ ratio and stress, indicating that the power ratio can serve as a distinguishing factor for assessing stress levels in brainwave data. The $\alpha$ to $\beta$ ratio was intended to establish a connection between the two most significant frequencies, facilitating the comprehension of an individual's cognitive state progression and enabling the observation of fatigue/tiredness development over a period [30]. Benjamin James Griffiths et al. [31] demonstrated that participating in a cognitive activity had a substantial impact on the $\alpha/\beta$ value in the domains of visual perception, auditory perception, and visual memory retrieval. Sanghee Kim et al. [27] employed the $\alpha/\beta$ method to examine the impact of modifications in architectural features (such as spatial aspect ratio, ceiling height, and window ratio) on relaxation arousal reactions. It follows that the $\alpha/\beta$ value can be chosen as an indicator for analyzing the relaxation–arousal response.

*1.3. Study Purpose*

The objectives of this study encompass three key facets (see Figure 1):

1. Examine the subjective perception disparities among individuals before and after the implementation of rural streetscape improvement.
2. Investigate the association between transformation components and the relaxation–arousal responses of brain electrodes.
3. Analyze the consistency between changes in subjective factors and changes in EEG $\alpha/\beta$ values.

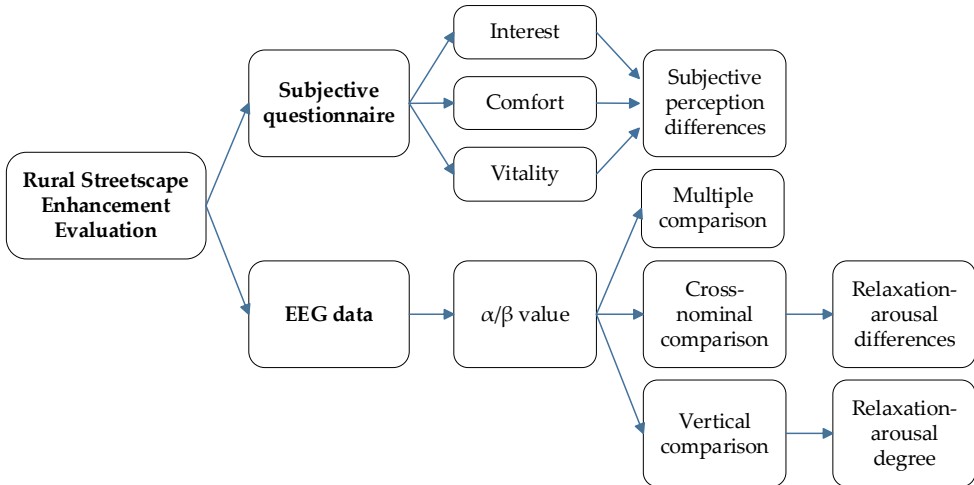

**Figure 1.** The proposed model.

## 2. Materials and Methods

### 2.1. Subject Selection

The experimental scene in question is situated in Shipo Village, specifically in Kangzhuang Township, Fuxing District, Handan City, Hebei Province. The main street of the village consists of five significant nodes (C1, C2, C3, C4, and C5), which are arranged in a north-to-south order. These nodes represent the current state of the scene. Additionally, five corresponding transformation scenes (T1, T2, T3, T4, and T5) have been selected as experimental images (see Figure 2). The experimental model of the status quo scene replicates the actual rural streetscape, while the retrofit scene model represents a retrofit plan tailored to the existing conditions of the rural streetscape.

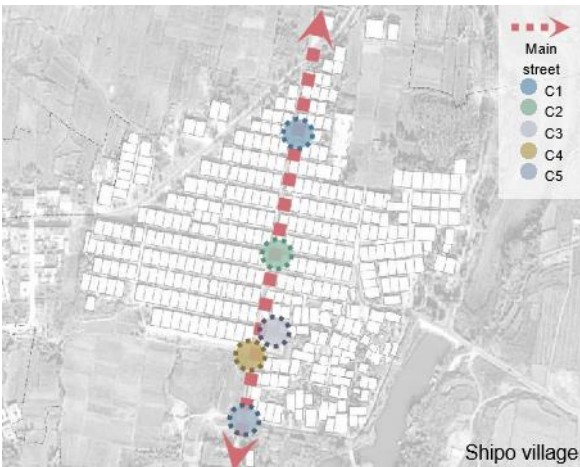

**Figure 2.** The location of the experimental prototype scene.

As seen in Table 1, the chosen scenes have been categorized into five distinct categories. C1 is a small public space node that currently suffers from neglect and lack of maintenance. T1 transformation strategies aim to enhance environmental quality by introducing greening measures, providing seating areas, and modifying the materials used in surrounding buildings. C2 represents the convergence of vertical and horizontal main streets in the rural area, serving as a densely populated meeting point for pedestrians. However, this node lacks architectural uniformity and fails to preserve the original rural landscape. The T2 retrofit procedures encompass the use of indigenous materials to substitute the initial building façade materials, the harmonization of the roof structure, and the augmentation of vegetation along the thoroughfares. C3 represents a node of activity space located along a secondary road adjacent to the primary street in a rural area. This node exhibits a noticeable

difference in elevation compared to the main road, and some of the residential houses within the node are abandoned, resulting in a visually unappealing landscape. In response to this, T3, the corresponding intervention, involves the removal of hazardous structures and the addition of a green park and landscape sketches to enhance the aesthetic quality of the area. C4 is situated at the terminus of the primary thoroughfare within a rural area. The primary enhancements implemented in T4 encompass the enhancement of the street frontage and the reconfiguration of the preexisting agricultural area to augment the variety of cultivated plant species and provide a more pronounced hierarchical structure within the agricultural domain. C5 serves as the central hub of a vast public green space, while T5 is designated to transform and arrange the existing untamed vegetation into an area suitable for cultivating crops and providing an immersive experience.

**Table 1.** Five groups of rural street scenes panoramic images.

| | Current Scenes | Transformation Scenes |
|---|---|---|
| The first set of scenes |  |  |
| | C1 | T1 |
| The second set of scenes |  |  |
| | C2 | T2 |
| The third set of scenes |  |  |
| | C3 | T3 |
| The fourth set of scenes |  |  |
| | C4 | T4 |
| The fifth set of scenes |  |  |
| | C5 | T5 |

The primary nodes of the village's main street in the current scenario are C1–C5, which are grouped in a sequential order from south to north. We aimed to develop a cohesive emotional feedback design to evoke emotions in locals or visitors as they go along this street due to its renovated ambiance. The experimental stimuli were arranged in a predetermined sequence.

The experiment employed Sketch Up 2019 software [32] to construct the initial 3D model, which was subsequently imported into Mars 2020 (Bright City Chongqing Technology Co., Ltd., Chongqing, China) to generate an image that could be experienced through a virtual reality device (HTC VIVE Pro, Wuhan Lingzhimiaojing Technology Co., Ltd., Wuhan, China). To enhance the fidelity and quality of the experimental setting, a VR scene map was employed, offering a 360° panoramic mode. This VR scene map comprised five distinct sets of visual stimulus elements. The VR panorama had a resolution of $2048 \times 1080$ pixels, commonly referred to as 2 K, and the visual stimuli were presented via VR glasses with a refresh rate of 60 Hz.

### 2.2. Selection of Questionnaire Indexes

To gather data on the environmental sentiments of participants across several situations, a questionnaire was designed. The questionnaire included three subjective indexes, namely Interest, Comfort, and Vitality. Participants were asked to assess these indexes on a five-point Likert scale, ranging from one to five. Previous studies have employed the following terms to evaluate urban public spaces: interest, which elicits sensations of enjoyment and attachment to the location; comfort, which is contingent upon the physical environmental factors present in the space, such as seating availability, street furniture, sidewalk width, trees, and so forth [33], as well as vitality, which gauges the level of diversity and liveliness within a given place [34].

To accurately determine the level of relaxation–arousal for the three indexes, we utilized a two-dimensional circular affective model (see Figure 3) based on Scherer's (2005) work. This model incorporates the original arousal–valence coordinates proposed by Russell (1979), which measure the degree of excitement/concentration and pleasure/sadness, respectively. Additionally, it included assessments for Control (low–high) and Conductive/Obstructive (favorable hindering). We can provide a more comprehensive description of the emotions in the model.

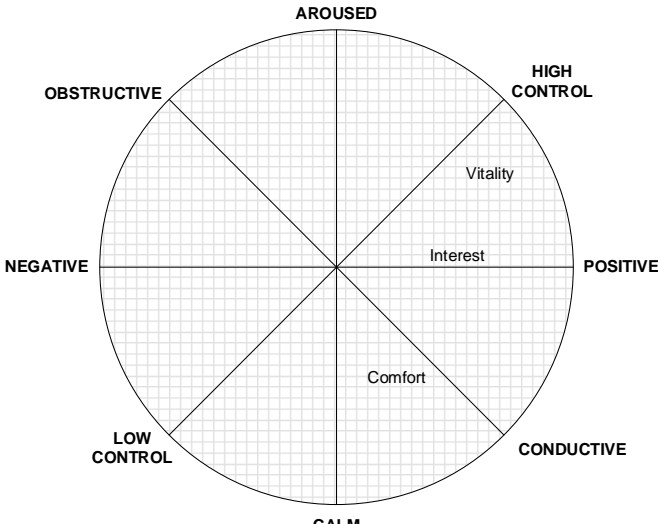

**Figure 3.** The position of Interest, Comfort, and Vitality in the model.

### 2.3. Method

The statistical analyses were conducted using SPSS 26.0 software, which was utilized to compute the mean EEG data for the five sets of scenarios. To assess potential disparities in subjective and objective data among various groups of comparison scenes, we will conduct

a normality test and analysis of variance on the $\alpha/\beta$ value o of the current group compared to the $\alpha/\beta$ value of the transformation group, as well as the Likert scale scores of the current group compared to the Likert scale scores of the transformation group. Subsequently, a comparative analysis will be conducted to examine the disparities in the reactions of the $\alpha/\beta$ value towards the rural streetscape both pre- and post-enhancement. Additionally, the EEG data visualization will be utilized to assess the relaxation–arousal responses and the extent of relaxation. The comparative analysis of the degree of involvement of areas or spots in the EEG topography map will be conducted after the elicitation of the associated emotions by the subjects. Lastly, we examined the relationship that existed between modifications in $\alpha/\beta$ values and modifications in subjective factor scores.

### 2.4. Experimental Process

The present study employed a voluntary recruitment method to enlist a sample of 30 Chinese university students, aged between 20 and 30 years. All participants possessed prior experience residing and pursuing education in rural areas and demonstrated a comprehensive understanding of rural environments. Sensitivity analysis for this sample size was conducted using G*power. This analysis was performed considering a *t*-test for within-subjects design—with two conditions (after viewing current scenes and after viewing transformation scenes), of 0.05, and a power of 0.8. Thus, our sample can detect the effects of medium or large size. Participants were invited to participate on-site, and a summary of their basic information is provided in Table 2. All subjects possessed satisfactory visual acuity or had their eyesight corrected and were devoid of any mental disorders to ensure accurate identification of visual stimuli. We aimed to examine the disparity in EEG response between the transformed scene pair and the present scene, as well as explore the emotional alterations induced by the present scene by designing the transformed scene to influence the present scene. Consequently, we selected participants who had no prior exposure to both the original scenario and the modified scene. Given that all participants were seeing the scenario we presented for the first time, their emotions were stimulated for the first time as well. Consequently, the EEG response signals collected were genuine and impactful.

**Table 2.** Participants' basic information (n = 30).

| Item | Details | Frequency | % |
|---|---|---|---|
| Gender | Male | 12 | 40 |
| | Female | 18 | 60 |
| Age | 20–25 | 24 | 80 |
| | 26–30 | 6 | 20 |
| Education background | Undergraduate | 2 | 6.67 |
| | Master | 28 | 93.33 |

Figure 4 illustrates the experimental methodology and setting employed in this investigation, along with the designated region for EEG measurements. The experiment was conducted in batches from 15 September to 25 September 2023, from 2:00 to 6.00 pm, due to the substantial number of participants. The studies were conducted within the Perceptual Engineering Laboratory, where the ambient temperature was consistently kept at 25°. Additionally, no external distractions were present during the trials. The experiments were conducted utilizing an EEG device as a measuring instrument. The EEG device recorded data from 32 channels at a sampling rate of 500 Hz. We characterize the placements of 32-channel EEG electrodes based on the International 10–20 method. The utilization of selective channels is a commonly employed preprocessing technique in the analysis of EEG signals. It reduces the use of too many unnecessary steps [35,36]. As the M1 and M2 electrodes are in the motor cortex, they are not considered to be part of the primary regions of the brain. POz electrodes are typically excluded from research investigations,

resulting in their eventual omission from analysis. EEG data were collected from 29 selected electrodes of the brain, including the frontal lobe (Fp1, Fpz, Fp2, F3, Fz, F4), left temporal lobe (F7, Fc5, T7, Cp5, P7), right temporal lobe (F8, Fc6, T8, Cp6, P8), central lobe (Fc1, Fc2, C3, Cz, C4), parietal lobe (Cp1, Cp2, P3, Pz, P4), and occipital lobe (O1, Oz, O2). The brain wave data were recorded using the SAGA64/32 portable EEG device from TIMS (https://www.tmsi.com/products/saga-32-64-128/ (accessed on 10 January 2024)).

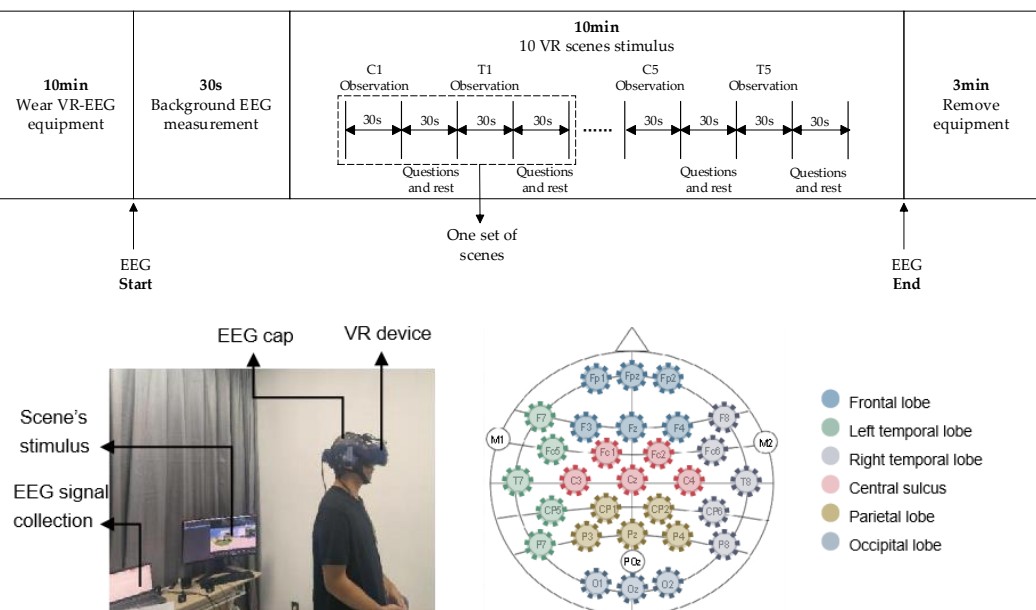

**Figure 4.** Experimental process and environment and EEG measurement area.

It is understood that short durations, like a 30 s window, are often used in psychological and cognitive research to assess immediate responses to stimuli, including those in VR environments. This timeframe is considered sufficient to observe and measure rapid cognitive and physiological responses to virtual stimuli [37].

Five sets of pre-existing visual stimuli were imported, resulting in a total of 10 scene graphs. Virtual reality glasses were utilized to present a panoramic slide presentation. Before commencing the experiment, pertinent demographic information was gathered from each participant. Additionally, participants were provided with detailed instructions about the experimental protocols and were aided in the proper application of the EEG and VR equipment. The experiment commenced by initially assessing the baseline background EEG measurements of each participant through 30 s of observing a blank room panorama. Subsequently, the participants were exposed to a series of VR scene slides sequentially. During this phase, the participants were instructed to maintain a stationary position, visually explore the scenes, and observe them for a duration of 30 s. After each scene, the participants were required to reorient themselves to their initial direction. Following this, the participants were presented with a 30 s blank room panorama and were asked to provide subjective evaluations by responding to three questions about interest, comfort, and vitality. Following the completion of the questions, participants were instructed to compose themselves and await further instructions before transitioning to the subsequent scenario.

Upon the conclusion of the trial, the participants' VR glasses and EEG devices were detached. The duration of the entire experiment was around 25 min.

## 2.5. EEG Signal Processing and Statistical Analysis

The EEG data do not exhibit a direct correspondence with an individual's relaxation–arousal response. To investigate the specific frequency band of interest, the EEG data must undergo preprocessing and extraction procedures. In the EEG data analysis, the data are first pre-processed, which includes filtering, artifact removal, re-referencing, and

feature extraction [38,39]. One of the benefits of eliminating artifacts is the reduction in or elimination of conflicting signals caused by muscular activity, such as facial expressions or head movements. This process improves the quality of EEG data and enhances the accuracy of the analysis. Then, statistical significance testing is used to identify characteristic values in brain regions involved in current scenes (C) and transformation scenes (T), as shown in Figure 5.

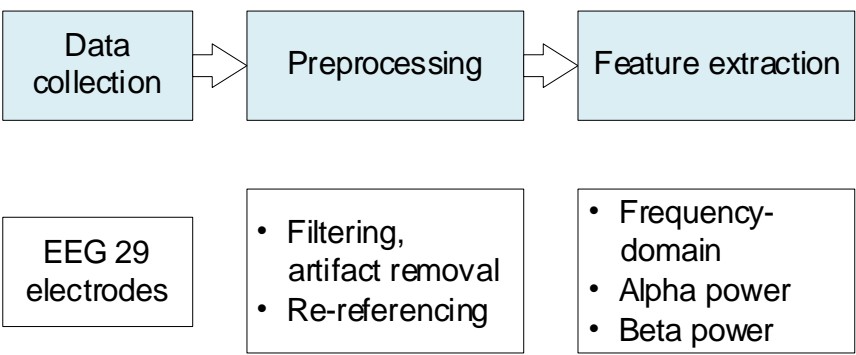

**Figure 5.** EEG signal analysis.

Because every person's EEG signal is different, we standardize the $\alpha/\beta$ value. We then do the reference acquisition—which also applies to the processing of the EEG signal— while the patient is not receiving any stimuli, and the resulting $\alpha/\beta$ value serves as the standard baseline.

$$\alpha/\beta_{\text{diff}} = |(\alpha/\beta)_{\text{n}} - (\alpha/\beta)_{\text{reference}}| \tag{1}$$

Equation (1) denotes the eigen value generated by the stimulus, the standard baseline, and the acquired difference, which is the variation in the individuals' emotional reaction upon seeing the stimulus.

## 3. Results

### 3.1. Questionnaire Results

The initial step was assessing the normality of the results for the three variables ("Interest", "Comfort", and "Vitality") in both the status quo and renovation scenarios within each group. The Shapiro–Wilk test yielded statistically significant findings, suggesting that the three index scores within each scene do not adhere to a normal distribution. To ascertain whether there was a significant difference in the scores of the three indexes between the current scenes and the transformation scenes, the Mann–Whitney U test was employed.

Figure 6 illustrated the distribution of data, as well as the mean scores and comparative differences for the three metrics of "Interest", "Comfort", and "Vitality" among the five sets of scenes (C1 vs. T1, C2 vs. T2, C3 vs. T3, C4 vs. T4, and C5 vs. T5). The data presented in the figure indicated notable disparities in the scores of the three indexes across various comparisons: C1 vs. T1, C2 vs. T2, C3 vs. T3, and C4 vs. T4 ($p \leq 0.01$). Additionally, there was a significant distinction in the scores of the "Comfort" index solely between C5 and T5 ($p < 0.05$). Furthermore, it was seen that the mean scores of the transformation scenes in each group surpassed those of the current scenes.

### 3.2. Multiple Comparisons of Transformation Scene (T1–T5)

To rule out the potential for individuals experiencing exhaustion, tiredness, and boredom after the trial (C4, T4, C5, T5), we made multiple comparison analysis tests of T1–T5. The $\alpha/\beta$ values of T1–T5 were subjected to a one-way ANOVA test. T1 and T2's $\alpha/\beta$ values differed greatly from T3, T4, and T5's values. There was a considerable difference between T3 and T5's $\alpha/\beta$ values. We obtained T1 < T2 < T3 < T4 < T5 by comparing the average $\alpha/\beta$ values of the transformation scenes (see Figure 7).

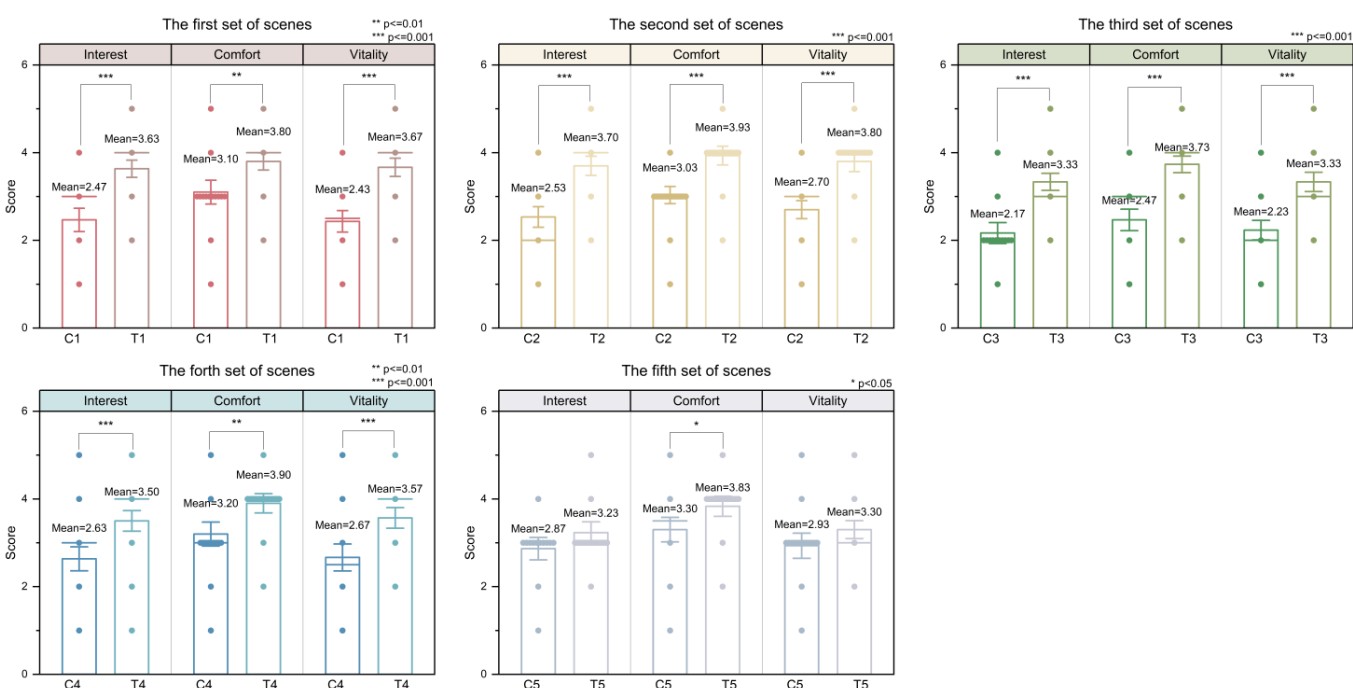

**Figure 6.** Questionnaire score distribution and T-test of five sets of scenes bar chart.

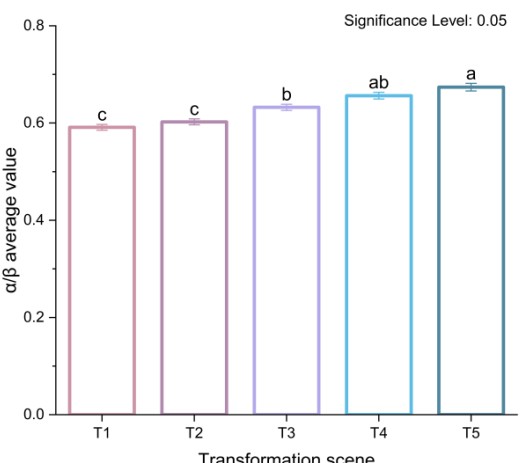

**Figure 7.** Multiple comparisons of transformation scene. (Significant differences between groups are indicated by different letters ($p < 0.05$)).

### 3.3. Differences in the Response of α/β Value to Rural Streetscapes

The initial step involved conducting a normality test on the $\alpha/\beta$ values across 29 electrodes for a sample of 30 subjects across 5 sets of scenes. The findings revealed that the data exhibited skewness. Consequently, the Mann–Whitney U test was employed to examine the disparity between the EEG data recorded during the status quo scene and the modified scene. Then, we identified the specific electrodes implicated in relaxation–arousal responses within the brain lobes.

Figure 8 presented an analysis of the distribution of $\alpha/\beta$ values, and the results of a differential comparison between C1 and T1 across 29 electrodes in the first set of scenes. Notably, a significant difference ($p \leq 0.01$) was observed in the $\alpha/\beta$ values of Pz electrodes situated in the parietal region when comparing the responses of C1 and T1. Notably, the $\alpha/\beta$ value was higher in T1 compared to C1, suggesting that the brain is in a state of relaxation. Furthermore, no statistically significant disparities were identified at any of the other electrodes. However, it was noteworthy that most of the electrodes

demonstrated greater relaxation responses during T1, except for the electrodes that did not exhibit substantial alterations.

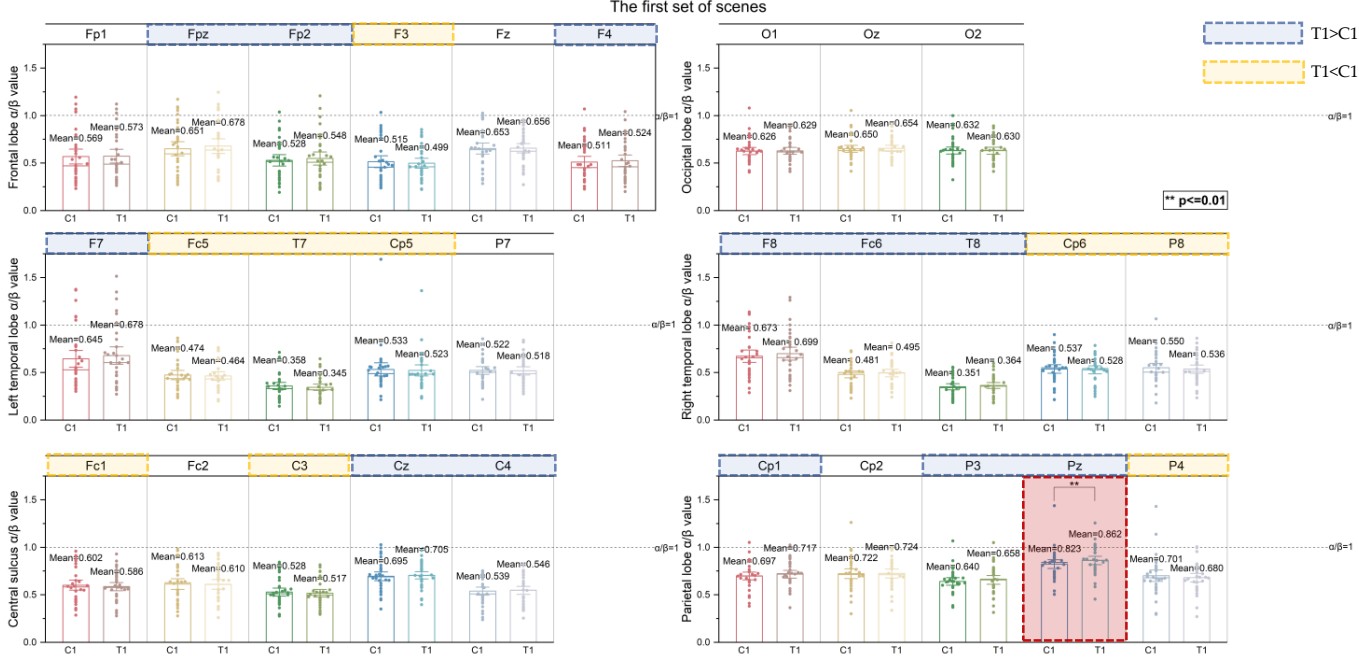

**Figure 8.** $\alpha/\beta$ value analysis results of the first set of scenes.

The findings from the analysis of the second set of scenes are depicted in Figure 9. Notably, a statistically significant disparity was observed in the $\alpha/\beta$ values of the T8 electrode situated in the right temporal lobe region ($p \leq 0.01$) and the Pz electrode located in the parietal lobe region ($p < 0.05$) between the response of C2 to T2.

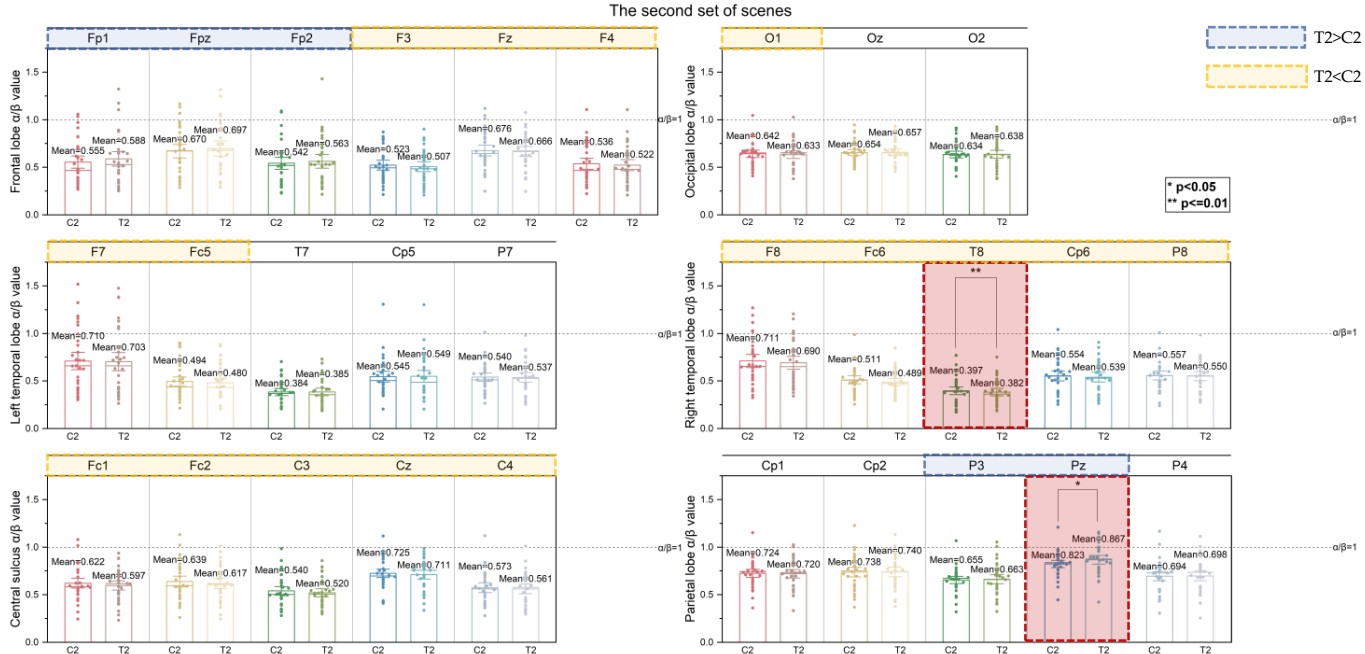

**Figure 9.** $\alpha/\beta$ value analysis results of the second set of scenes.

The T8 electrodes exhibited a decreased $\alpha/\beta$ value in T2 in comparison to C2 and demonstrated higher arousal responses in the right temporal lobe. The $\alpha/\beta$ value of the Pz electrode on T2 exhibited a greater average compared to that of C2. This finding aligned with the performance observed in the scene group. Furthermore, among the other

electrodes that did not show significant differences, all electrodes showed higher arousal responses except for the Fp1, Fpz, and Fp2 electrodes in the frontal region and the P3 electrode in the parietal region, which showed higher relaxation responses due to the stimulation of T2.

The findings from the examination of the third set of scenes in the study are presented in Figure 10. Notably, the analysis revealed a statistically significant distinction ($p < 0.05$) in terms of the activity observed in the Cp5 electrode located in the left temporal lobe region, as well as the P3 and Pz electrodes situated in the parietal lobe area.

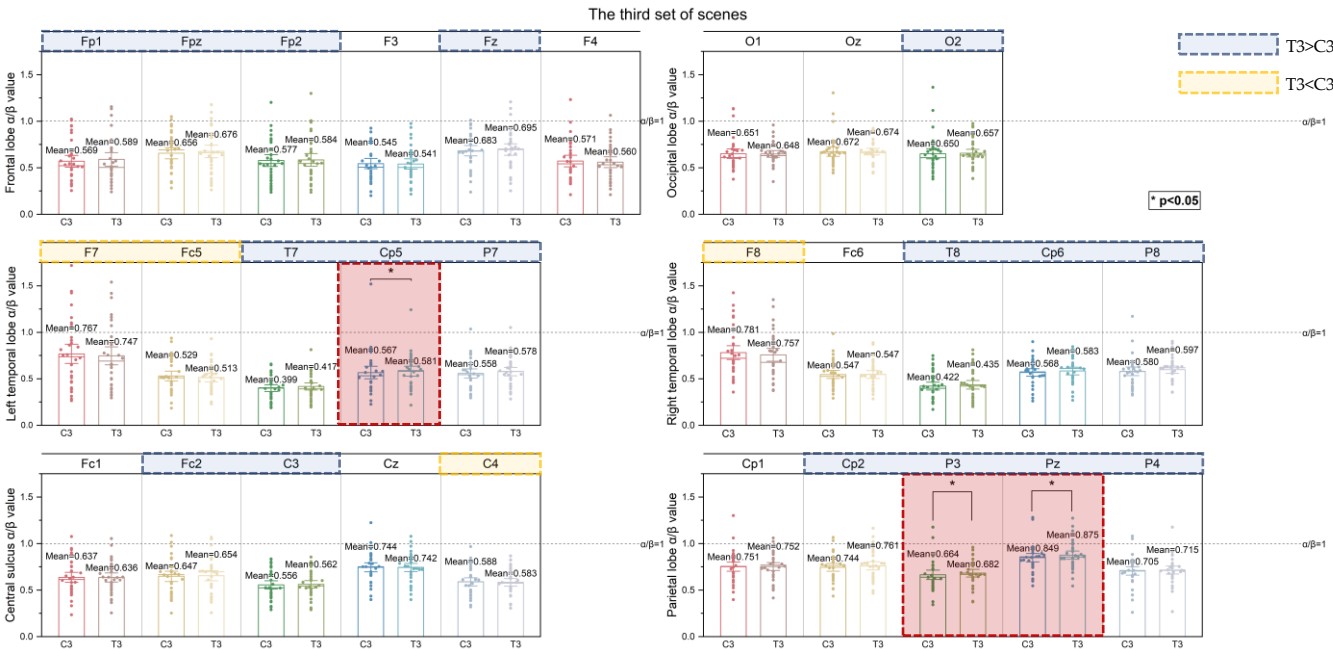

**Figure 10.** $\alpha/\beta$ value analysis results of the third set of scenes.

The $\alpha/\beta$ value on T3 was higher in the Cp5 electrode located in the left temporal lobe area. The $\alpha/\beta$ values of P3 and Pz electrodes located in the parietal area on T3 exhibited higher averages compared to those of C3 electrodes. Additionally, in significantly different electrodes ($p < 0.05$), a comparison of the number of electrodes that produced more relaxation responses in the first, second, and third sets of scenes showed that only Pz electrodes in the first and second sets of scenes exhibited relaxation responses, whereas the number of electrodes that produced relaxation responses was greater in the third set of scenes.

The findings from the examination of the fourth set of scenes are presented in Figure 11. Among these, the $\alpha/\beta$ value of F4 electrodes in the frontal lobe region exhibited a statistically significant distinction in the reaction between C4 and T4 ($p < 0.05$). T4 had more relaxation responses than C4 with a higher mean $\alpha/\beta$ value. Furthermore, it was worth noting that there were no statistically significant changes seen on any of the remaining electrodes. However, most of the electrodes exhibited a greater number of relaxation responses on T4.

As depicted in Figure 12, the electrodes that displayed notable variations in $\alpha/\beta$ values were observed to be more widely distributed and more abundant in the fifth set of scenes compared to the groups of situations. Significant differences were observed in the response of the Fc5 electrode located in the left temporal lobe area; the T8, Cp6, and P8 electrodes situated in the right temporal lobe area; and the P4 electrode in the parietal lobe area ($p < 0.05$). Moreover, T5 had more relaxation responses than C5 with a higher mean $\alpha/\beta$ value.

Furthermore, among the remaining electrodes that exhibited no substantial disparities, it was observed that a greater number of electrodes exhibiting relaxation responses were

present in temporal lobe areas. Conversely, the alterations in the $\alpha/\beta$ values within the frontal and occipital lobe regions were comparatively less prominent.

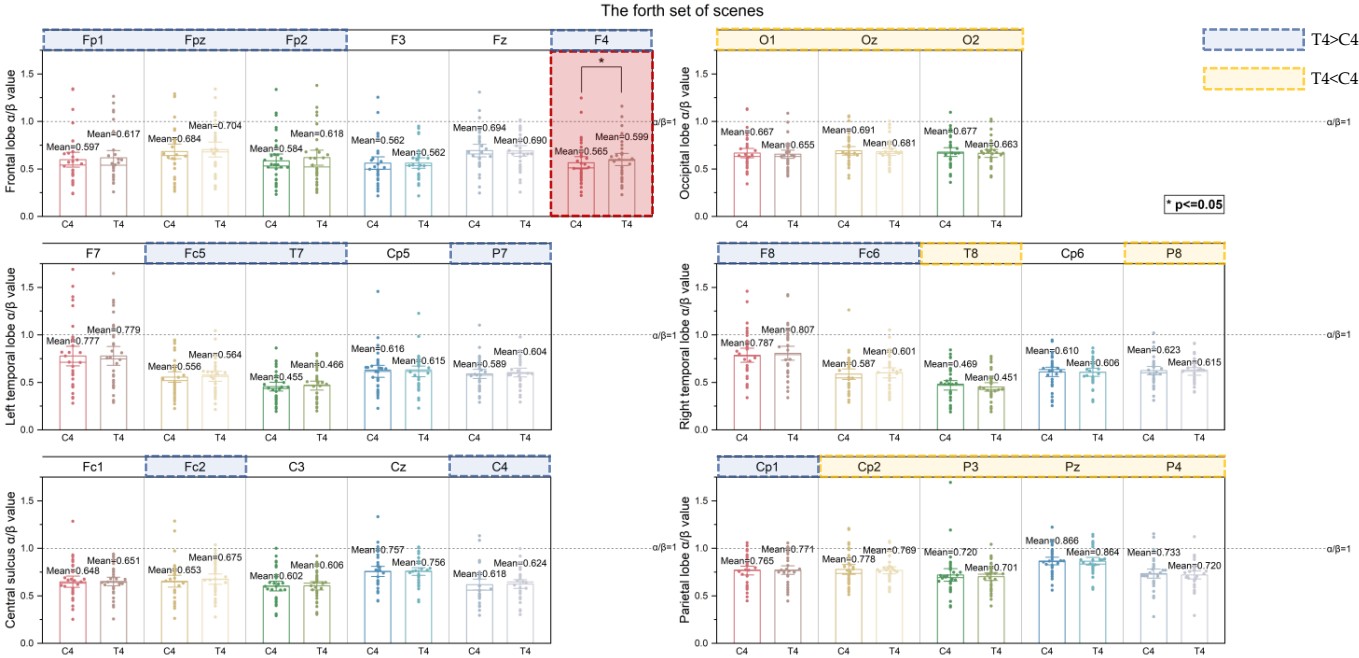

**Figure 11.** $\alpha/\beta$ value analysis results of the fourth set of scenes.

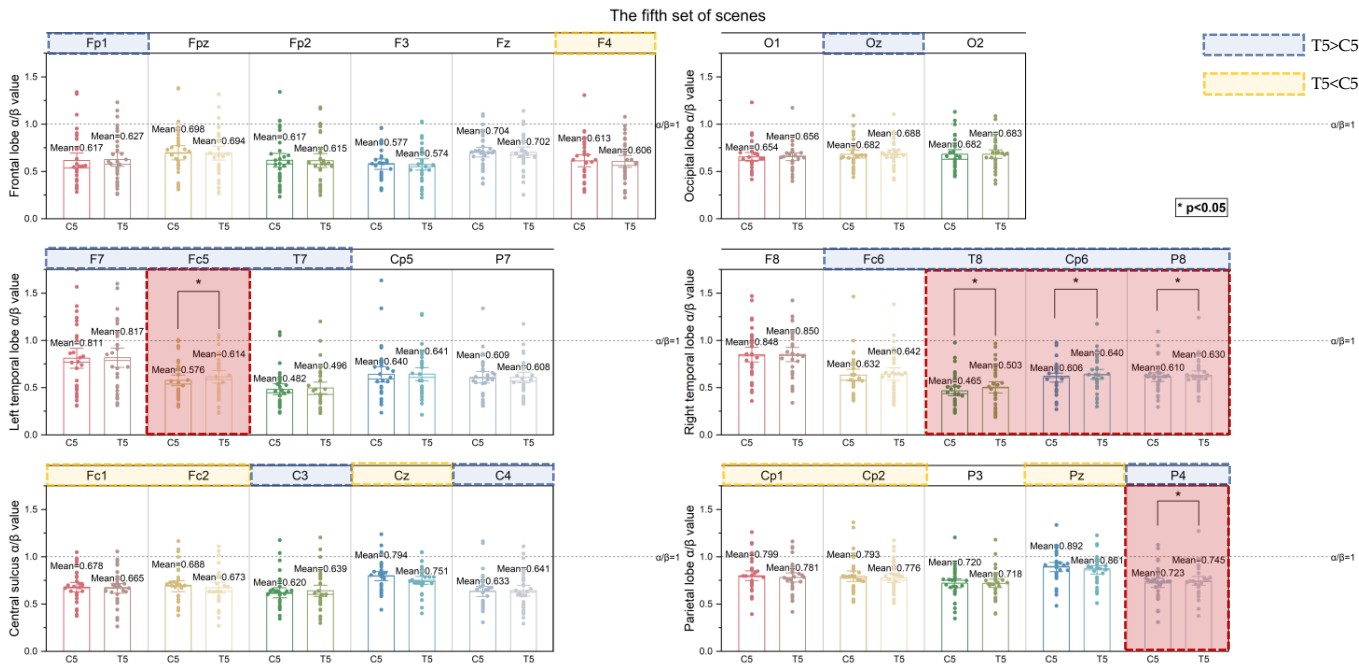

**Figure 12.** $\alpha/\beta$ value analysis results of the fifth set of scenes.

### 3.4. Comparative Analysis of Relaxation–Arousal Degree of Pz and T8 Electrodes

The Pz and T8 electrodes were the ones that differed significantly across multiple sets of scenes and could therefore be used for group-to-group comparisons of relaxation–arousal degree. The data above indicate that when it came to significant variations ($p < 0.05$) in the values of the $\alpha/\beta$ indicator before and after the rural streetscape augmentation, the Pz electrode in the parietal area outperformed the other electrodes, as depicted in Figure 13a. The Pz electrodes exhibited higher relaxation responses in T than C across three scene groups, namely the first set of scenes, the second set of scenes, and the third set of scenes.

The highest amount of relaxation was observed in the second set of scenes, followed by the first set of scenes, and lastly the third set of scenes. In Figure 13b, it can be shown that the T8 electrode, located in the right temporal lobe region, had a higher arousal response in T than C of the second set of scenes, but it displayed a higher relaxation response in T than C of the fifth set of scenes.

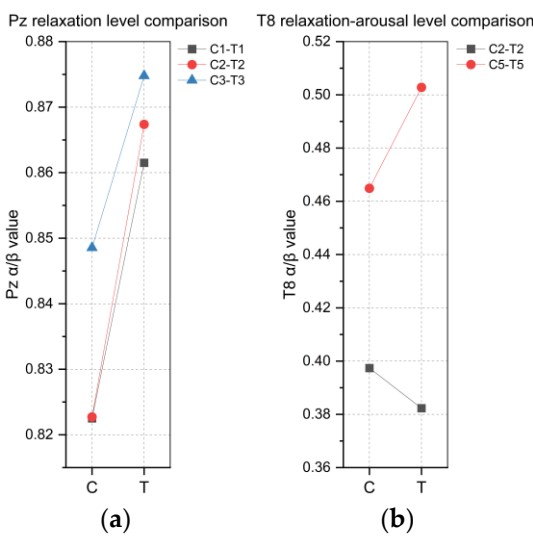

**Figure 13.** Comparison of relaxation–arousal degree. (**a**) Comparison of relaxation degree of Pz $\alpha/\beta$ value; (**b**) Comparison of relaxation–arousal responses of T8 $\alpha/\beta$ value.

### 3.5. Trend Analysis of Changes in Subjective Scales and $\alpha/\beta$ Values

Figure 14 illustrates the patterns seen in the average scores of three subjective indicators, namely Interest, Comfort, and Vitality, as contrasted to the trends in the average $\alpha/\beta$ values on the electrodes exhibiting significant distinctions. The results indicated that variations in the $\alpha/\beta$ values of Pz electrodes in the first, second, and third sets of scenes were positively associated with variations in the three subjective factor scores, while variations in the $\alpha/\beta$ values of T8 electrodes in the second and fifth sets of scenes were negatively associated with variations in the three subjective factor scores. The $\alpha/\beta$ values of Fc5, Cp6, P8, and P4 electrodes in the fifth set of scenes were also positively correlated with the changes in the scores of these three subjective factors. The changes in the $\alpha/\beta$ values of Cp5 and P3 electrodes in the third set of scenes were positively correlated with the changes in the scores of Interest, Comfort, and Vitality.

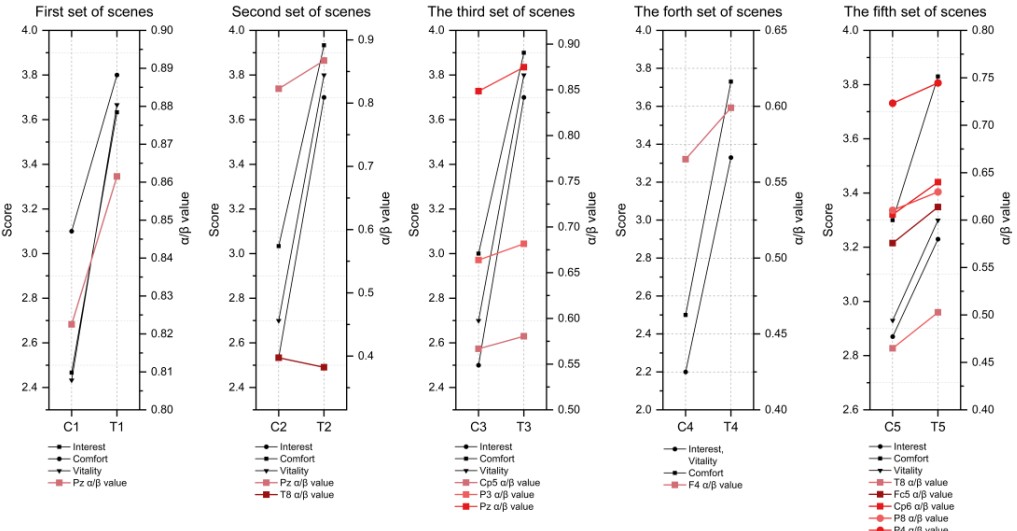

**Figure 14.** Graphical comparison between average scores and $\alpha/\beta$ value for five sets of scenes.

## 4. Discussion

The objective of this study was to assess the impact of rural streetscape enhancement on individuals' relaxation–arousal responses. This was achieved by measuring the cognitive stimulus responses obtained from the EEG in a virtual reality environment. Additionally, the study aimed to investigate the specific elements of streetscape modification that may have a beneficial effect. The findings of this study indicate that there were noticeable alterations in the subjective evaluation scale and $\alpha/\beta$ values when participants were exposed to the rural streetscape.

(1) Significant subjective cognitive differences existed between subjects' perceptions of current scenes and transformation scenes in the different groups. Furthermore, the transformation scenes exhibit enhancements in three subjective indices ("Interest", "Comfort", and "Vitality"). These findings suggest that the transformation scenes possess a greater capacity to elicit positive emotions. The transformation scenes were perceived as more aesthetically appealing in terms of overall perception, hence potentially eliciting more enjoyable and positive emotional responses. Based on the placement of the three subjective indexes on the two-dimensional emotion model, it was observed that the subjects' arousal responses in T1, T2, and T4 exhibited the most significant variation in their overall perception of the environment. Conversely, the subjects' relaxation responses in T3 and T5 constituted the predominant portion of their emotional experiences.

(2) The multiple comparisons among T1–T5 confirmed our initial hypothesis that human relaxation levels were higher when exposed to the transformation scenes on the main street of the hamlet.

(3) Through a longitudinal comparison of the Pz electrodes, we observe that the enhancement measures commonly implemented for the three categories of scenes primarily involved the augmentation of vegetation and the integration and synchronization of architectural facades along the street. The differences were the green area and the scale of the space the subjects were in. Neuropsychological research indicates that the parietal region of the brain is principally linked to spatial attention and spatial memory [40]. The Pz electrode in the parietal lobe exhibited a greater number of relaxation responses on T1. This implies that the augmentation of centralized green spaces and open spaces, as well as the integration of architectural components and shapes along the roadway, can potentially boost levels of visual attention and cognitive processing. Prior studies have demonstrated that the presence of plants in a street environment has a notable and favorable impact on several human emotions, such as feelings of comfort and vitality [41]. These findings align with the results of the present study. The Parietal zone Pz electrodes produced relaxation responses in both T2 and T3. These responses were characterized by the presence of various plant species, the creation of green parks, and the improvement of building facades as part of the remodeling process. Other studies have also demonstrated that a street scene that is more diverse and less fragmented leads to positive emotional experiences [16]. Furthermore, the observed variations in relaxation degree across the three groups further indicate that distinct street scales have an impact on individuals' pleasant moods, which is like previous research demonstrating that the scale of indoor spaces can elicit either good or negative physiological reactions in users [27].

Based on the longitudinal analysis of T8 electrodes inside the right temporal lobe region, both the second and fifth sets of scenes share a common characteristic in that they are situated at street intersection nodes. However, the restoration of the previous architectural form to the area is the most apparent visual alteration observed in the second set of scenes. This results in a cohesive and harmonized aesthetic along the facades of the street buildings. On the other hand, the fifth set of scenes distinguishes itself through the planning of the original farmland as a crop planting and experience area. We planted country crops as landscape plants to resemble the country scene more closely. In this investigation, the sole arousal response observed was in the right temporal lobe region T8 electrode at T2. Notably, T2 was situated at the intersection node and is encompassed by architectures. In contrast, T5 is situated near a street intersection node; however, it encompasses a substantial

expanse of green open space on its western side. Therefore, T2 exhibits the most limited line of sight in comparison to other transformation scenes. The temporal lobe regions are responsible for several visual functions, including spatial orientation, stereoscopic vision, and color perception [42]. Hence, the presence of architectural structures in the surroundings enhanced the participants' levels of attentiveness and cognitive attention. The primary features of transformation in T5 exhibited similarities to the above scenes, thereby eliciting a relaxing response.

In addition, there existed more electrodes producing relaxation responses in T3 compared to T1 and T2, and the difference between these three changes was the difference in green areas (T1 was small green area remediation, T2 was green planting remediation along the road, and T3 was the addition of green parks), which suggests that remediation measures to increase the extent of green areas and landscape vignettes have a positive relationship with relaxation responses. T5 exhibited the highest number of relaxation response electrodes in the study, this finding suggests that landscaping and remediation of large public green space nodes can also be effective in enhancing relaxation responses.

(4) Vecchiato et al. [43] conducted a study that established a correlation between self-reported emotional responses and EEG responses. The researchers proposed that utilizing emotion identification methods that rely on vital sign readings could serve as an objective tool for spatial assessments. In the current investigation, an examination was conducted to assess the congruence between subjective index scores and EEG data trends. The results revealed the convergence between subjective factor ratings and changes in $\alpha/\beta$ values on the electrodes was not seen. The $\alpha/\beta$ values of the T8 electrode located in T2, the Cp5, P3, and Pz electrodes situated in T3, the T8 electrode in T4, and the T8, Fc5, Cp5, P8, and P4 electrodes positioned in T5 exhibited congruence with the subjective questionnaire. The limited number of indications present in the subjective evaluation scale may not adequately capture the extent of correlation and consistency between the scale and the EEG data.

(5) This study, however, possesses certain drawbacks. Initially, the participants in the study consisted of college students, rather than the primary demographic of rural inhabitants. Consequently, it is important to acknowledge that there may exist certain disparities between the findings obtained and the real-world circumstances. Furthermore, the VR technology employed in this study cannot replicate certain sensory pathways seen in real-life scenarios, such as touch and hearing. Additionally, the VR simulation used in this experiment does not fully replicate the authentic environment, leading to potential inaccuracies in the results.

## 5. Conclusions

This study investigated the alterations in $\alpha/\beta$ values in the EEG of participants exposed to five different sets of rural streetscape scenes using VR-based immersive spatial experience. A total of 30 subjects were included in the study. The study sought to explore the factors that may impact relaxation–arousal responses to these modifications. The findings of the study indicate that the Pz electrodes located in the parietal area exhibited heightened sensitivity towards visual stimuli associated with the green landscape and the architectural features of building facades along the street. There was a favorable correlation observed between the extent of green space, the integration of green elements, and the integration of architectural elements with relaxation responses. The presence of properly proportioned streets elicited relaxation responses, while the relaxation responses of respondents were diminished by empty, monolithic types of public space. The electrode located in the right temporal lobe region, specifically T8, demonstrated the highest level of sensitivity towards the overall perceptual response to the surrounding environment. The presence of a built environment in proximity evoked an arousal response and enhanced the concentration and focus of the participants. Conversely, when the site provided an unobstructed view of an open field, a sense of relaxation was elicited.

Moreover, it was observed that the subjective scale findings did not precisely align with the $\alpha/\beta$ values, yet certain resemblances were identified. Therefore, it is not advisable to

exclusively rely on self-reported psychological assessments when evaluating environmental perceptions, emphasizing the need for further investigation into the correlation between subjective and physiological assessments in future research.

The consideration of user perception is frequently overlooked during the design phase of rural habitats. The research contributes to the identification of individuals' emotional requirements and preferences about environmental aspects in rural settings. Additionally, they offer valuable insights for designers in determining the most favored and habitable rural public areas for residents. Simultaneously, there will be a greater exploration of perceptual studies in the domains of architecture and rural areas in the future.

**Author Contributions:** H.R.: conceptualization, methodology, supervision, review, editing; Y.W.: conceptualization, methodology, software, validation, formal analysis, investigation, resources, data curation, writing, visualization, project administration; J.Z.: conceptualization, methodology, supervision, review, editing, funding acquisition; Z.Z.: methodology, data curation. Q.W.: supervision, review. All authors have read and agreed to the published version of the manuscript.

**Funding:** This research was funded by the Hebei Provincial Social Science Fund through Research on the Risk Assessment System of National Spatial Planning Based on Sudden Public Health Incidents (HB20GL055).

**Institutional Review Board Statement:** The study was conducted according to the guidelines of the Declaration of Helsinki and approved by the Institutional Review Board of Hebei University of Engineering (protocol code BER-YXY-2023031, approved 10 June 2023).

**Informed Consent Statement:** Informed consent was obtained from all subjects involved in the study. Written informed consent has been obtained from the subjects to publish this paper.

**Data Availability Statement:** The data presented in this study are available on request from the corresponding author. The data are not publicly available due to the large model files and large at volume.

**Conflicts of Interest:** The authors declare no conflicts of interest.

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
