# Peer review of "Evaluation of the Impact of VR Rural Streetscape Enhancement on Relaxation–Arousal Responses Based on EEG"

_applsci, doi:10.3390/app14072996_

Round 1

Reviewer 1 Report

Comments and Suggestions for Authors

The presented work is very interesting. The research part and the way of its presentation should be appreciated. Selected comments for possible supplementation.

1. In the study part, the impact of the size of the research population on the representativeness of the obtained results was not widely recognized. Please complete this.

2. Was the origin of the students subjected to the study analyzed? Place of residence: urban/rural and the influence of this factor on research results?

3. Could the place and time of the study have influenced the results?

Reviewer 2 Report

Comments and Suggestions for Authors

This study investigated subjects’ relaxation-arousal responses using EEG while observing different street scenes using VR. The purpose was meaningful and the methods were innovative. However, the description of the aims was not clear, and the data analysis was untenable, especially the assumption of normal distribution. Additionally, the explanations and discussion of the results were far-fetched. Please carefully interpret the meanings of comfort/vitality/interest/relax/happy. Last, the English needs to be improved. The following detailed comments might help the authors to improve this manuscript.

Abstract

1.       The English is difficult to be understood. Especially the first sentence and the second finding. For the second finding ‘Every transformation scene..., except for the T8 electrode...’, is the T8 electrode a transformation scene?

1. Introduction

2.       The authors mentioned the aim/objective/ goal many times in this manuscript. First, since there is a subsection named 'study purpose' (section 1.3), there is no need to mention them again at the end of sections 1.2 and 1.3. Second, even in section 1.3, the content of lines 124-130 was similar to lines 130-137, please avoid repetition. Lastly,  please mention the facets directly after 'The objectives of this study encompass three key facets:", without mentioning 'this study aims to...' and 'the objective of this study is...' again.

3.       Line 67, ‘...design elements including shape, layout, materials, scale, and color have been found to positively correlate with emotional reactions in neuro-scientific experiments’, what do you mean by ‘positively correlate’? Can these variables be quantified?

4.       Line 92, what do you mean by ‘a good understanding of the rural environment’?

5.       Line 129, please add related references.

6.       For Figure 1, please carefully check the logic and connections of these elements, and definitions of comfort/interest/vitality/cognitive. Cognition is a mental process of understanding knowledge, how could a comfort level be used to test cognitive differences? Besides, what is the difference between relaxation-arousal reaction and relaxation-arousal level in this study?

2. Materials and Methods

7.       Line 147, please add the number after ‘see figure’.

8.       For Figure 3 and line 202, ‘we included assessments for Control (low-high) and Conductive/Obstructive (favorable-hindering) to provide…’, did you add them? They were already included in the original Scherer’s circumplex model. Additionally, please check the locations of interest and vitality.

9.       Lines 214-223, this paragraph should be moved to the introduction section or removed...Please just describe the methods used in the current study in this section.

10.   Line 252-254, please rephrase this sentence, it’s unclear.

11.   Line 268, please add the reference in the bibliography and change the name of the article to a number. Similar to lines 297-299.

12.   Figure 5, ‘artifact removal’ should be one phrase. Styles should be consistent. The last box is not relevant to the statistical significance testing.

13.   Line 312, change ‘--Formula 1’ to  (1), besides, change ‘formula’ in line 313 to ‘equation’.

14.   Please keep the names of the same parameters the same in the whole paper, for consistency. For example, the α to β ratio index (line 128), α to β ratio (line 214), the α/β ratio (line 220), the α/β (line 226), the α and β ratio (line 319), the alpha and beta ratio index(line 323), the α and β ratios (line 329), the α/β index values (line 361), α/β index (line 368), α/β index value (line 403), α/β indicator values (line 448), α/β values (line 453)… are they the same?

15.   Line 328-329, can comparative analysis used to examine correlations between two continuous variables?

3. Results

16.   Line 343, ‘To ensure consistent analysis of the data, it is assumed that the results of the questionnaire a normal distribution’, this is questionable! You couldn’t ignore the Shapiro-Wilk and the skewness tests and assume they are normally distributed. If you ignore the test results, then what's the purpose of conducting the test? More importantly, you cannot assume they followed a normal distribution while they didn't, it’s unprofessionally. Please use the Mann-Whitney U test instead of the t-test.

17.   Line 346, ‘between the scores of the three indexes for both the current group and the transformation group’, please carefully check what did you compare? Between which two groups? It should be ‘difference in the scores of the three indexes between the current group and the transformation group’. Similar comment for lines 387-389, it should be ‘in the α/β values of the T8 electrode....between the response of C2 and T2’. Please check other descriptions.

18.   Line 360, ‘adjustments were made for repeated comparisons including T1-T5’, what are the adjustments? please introduce them in detail in the methods section.

19.   Figure 7, what do the letters (a, b, c) mean?

20.   Lines 393-395, first, the first Fp2 should be Fpz. Second, what do you mean by 'exhibiting notable variations in α/β values in response to relaxation'? it's not clear. What’s the definition of ‘notable’ in this study? All the electrodes exhibited variations when the scenes changed from C2 to T2. Do you mean the α/β values of these electrodes for T2 were higher than C2?? if so, similar results were observed for P3 and Pz. Why did you say ‘the remaining electrodes demonstrated an arousal response’?

21.   Line 406, what do you mean by 'elicited relaxation responses in the third set of scenes'? Do you mean the α/β values were higher for T3 than C3, or the α/β values were generally higher for all the scenes (both T3 and C3)? If it's the first, then it's not opposed to the first sets of scenes (see line 381, 'it is noteworthy that most of the electrodes demonstrated greater relaxation responses during T1'). If it’s the second, then please mention the threshold of the α/β value to distinguish the ‘relaxation’ and ‘arousal/alertness’, and provide related references.

22.   Line 412, ‘The average a/ß index value was greater than that of C4’, the value of what was greater than that of C4? please be precise.

23.   Line 424, please keep this description (p<=0.05, p≤0.05) the same in the whole manuscript. Besides, it’s better to provide the exact p-values.

24.   Lines 424-425, did you make the same comparison for other sets of scenes? If so, please mention the corresponding results. It’s better to keep the analyses and related descriptions the same for the five sets. Besides, what does the ‘greater’ mean? Was the difference significant? If so, please mention the p-value.

25.   Line 427-429, how did you get this conclusion? What do you mean by 'displayed relaxation responses'? do you mean they displayed higher α/β values for both T5 and C5 scenes? if so, it seems the values of Frontal and occipital lobes were even higher...or do you mean the α/β values for T5 were higher than C5? if so, please rephrase this sentence to make it clear. Additionally, what is the threshold of the α/β values to represent the ‘relaxation response?

26.   Section 3.4, why did you emphasize ‘Pz and T8 electrodes? Besides, the information shown in this subsection has already been introduced in the previous subsection.

27.   Lines 436-438, please be precise! ‘The Pz exhibited higher relaxation responses in T than C…’.

28.   There are too many figures in this manuscript, some of them could be changed to tables.

29.   Section 3.5, the discussion in this section is too far-fetched. According to your description, it seems that 'comfort' is the opposite of 'vitality' and 'interest', since comfort corresponds to the relaxation response (line 458) while the other two indicators contradict the relaxation response (line 453,456). Please make sure the definitions of comfort, interest, and vitality... they are three distinct concepts. Moreover, the relaxation cannot be seen as emotion (line 457).  Please carefully choose the words and interpret the results.

4. Discussion

30.   Line 478, ‘Substantial variations in .. exist between individuals' perceptions’, how did you get this conclusion? Did you check the difference between different subjects? No related results were mentioned in the ‘results’ section.

31.   Line 489, did you ask about enjoyment? which variable is related to enjoyment? Besides, this description is not consistent with the results shown in section 3.2 (line 362: The results revealed significant differences in the α/β values between T1 and T2 compared to T3, T4, and T5). Please be precise.

32.   Line 516, is there any real connection between reference 27 and the current one??? Of course, different spaces could elicit different reactions…

33.   Line 522, ‘enhances the diversity of plant species’, however, only crops were planted in T5, right? How could this enhance diversity?

34.   Line 529, ‘Weren't they all situated at the intersection nodes? (line 517 'scenarios share a common characteristic in that they are situated at street intersection nodes')

Line 539, ‘smallest green space area and elicited most relaxation response’, how could this suggest a 'positive correlation between the size of green area and the level of relaxation' Please check the logic!

35.   Line 542-544, Vecchiao et al. examined the emotional responses, did you check any emotion-related variables? these results are not comparable.

5. Conclusion

36.   Line 554, did you investigate emotional responses?

37.   Line 565, how could you get this conclusion related to ‘happy mood’? Did you investigate happy mood, sensitivity, or concentration in this study?

Line 572, don't cite other studies' findings in the conclusion. Please only focus on your findings in the conclusion.

Lines 578-584, please move this to the discussion.

38.   Line 589, please move the last sentence to the beginning of this section, and mention the contribution and implication of this study.

Comments on the Quality of English Language

The quality of English have to be improved. 

Reviewer 3 Report

Comments and Suggestions for Authors

The originality of the authors’ approach is in their contribution to exploitation of public participation in the design process with the use of modern technologies like Virtual Reality.

In contrast to so called “Digital Twins” related mostly to factories and warehouses, the authors have focused on “city and rural areas” that actually affects regular citizens. The underlying idea is to provide sound evaluation needs and implications for VR environment that probably be important part of Metaverse.

Professionally prepared  questionnaires (subjective) and objective  corroborative  (EEG) evaluations have been prepared to confirm the authors’ hypotheses. There are a number of already available methods of user experience evaluation based on  various design criteria and the authors refer to typical assessment of factors like space metaphors, image-objects and perception of environments. The basic line for the analysis is the Scherer's model (depicted in Figure 3.) However, there are other models worth comparison/evaluation (like  Mechanics Dynamics Aesthetics MDA or Action Gameplay Experience AGE, not to mention ‘classic’ Mihaly  Csíkszentmihályi’s Theory of Flow) that prove to work well with the user’s engagement in interactive environments. I also suggest looking closer at the Perceived Experience Questionnaire, ITC-SOPI (Sense Of Presence Inventory), MEC Spatial Presence Questionnaire, Novac and Biocca  Questionnaire (Lessetier J. et al., Development of a New Cross-Media Questionnaire, 3rd Int. workshop of PRESENCE, 2000 )

VR is characterized by distinctive feature of being totally ‘immersive’, where user is immersed in the virtual scenery. Maybe populating the scene with humans-citizens helps to make it more casual and ‘cozy’ compensating some scale distortions and artificiality of experience (the effect of “abandoned city”). According to the paper people looked primarily for comfort that is a highly subjective feature closely related to texture and lighting quality and is extremely hard to obtain in purely synthetic VR (hence the popularity of Cinematic VR or 360 Videos that bring the viewer to filmed real environments, there are plenty of them on Youtube 360… )  

Round 2

Reviewer 2 Report

Comments and Suggestions for Authors

Thank the authors for patiently answering all the comments. Most of them were addressed in the revised manuscript. However, there are still some sentences that are not clear. To make it easier to be followed by the readers, some improvements still need to be made. The English still needs to be improved. The following detailed comments could help the authors to further revise the paper. 

1.       Please check the tense of the verbs. Please use past tense in the Results section and when referring to previous studies, because these findings and outcomes have already been obtained and observed during the research or by previous studies.

2.       Please check the reference style.  If the names of the authors were mentioned, then please add the number after the authors' names.

Abstract

3.         ‘Every transformation scene(T1-T5) showed …, except for the T8 electrode’. This sentence is still confusing. It sounds like T8 is one of transformation scene (T1-T5), however, T8 is not included in T1-T5. Actually, they are totally different things, and can’t be compared. You can say ‘Every electrode showed…, except for T8 …’ Please check the usage of 'except for... 

Introduction 

4.       For section 1.3, the authors didn't address the related comment successfully.

5.       First, since there is a subsection named 'study purpose' (section 1.3), there is no need to mention them again at the end of sections 1.2. Second, please mention the facets directly after 'The objectives of this study encompass three key facets:", without mentioning 'this study aims to...' and 'the objective of this study is...' again.  For example, 'objectives of this study encompass three facets: 1. examination the subjective perception disparities....; 2, investigation the correlation between ....'.

Results

6.       Figure 3.7, ‘Marked with different letters indicate significant differences (p<0.05)’, there is a grammar issue. Please rephrase it.

7.       Lines 373-375, ‘except for the frontal area Fp1, Fpz, and Fp2 electrodes and the parietal area P3 electrode, …, all other electrodes that did not exhibit significant differences displayed higher arousal responses in comparison to C2’, does this mean Fp1, Fpz, fP2 exhibited a significant difference between T2 and C2? However, they did not, according to Figure 9.

8.       Line 386, ‘compared to the first and second sets of scenes where only Pz electrodes produced a relaxation response,’ T8 electrodes also produced a relaxation response in the second sets, didn't it?

Discussion

9.       Line 513, ‘a positive correlation between the size of the green area and the level of relaxation response among the participants’, How many different sizes of green areas did you investigate? Please be precise!

Comments on the Quality of English Language

English must be improved.
